# Transient and Long-Term Improvements in Cognitive Processes following Video Games: An Italian Cross-Sectional Study

**DOI:** 10.3390/ijerph19010078

**Published:** 2021-12-22

**Authors:** Rosa Angela Fabio, Massimo Ingrassia, Marco Massa

**Affiliations:** Department of Clinical and Experimental Medicine, University of Messina, 98121 Messina, Italy; mingrassia@unime.it (M.I.); dott.marcomassa@gmail.com (M.M.)

**Keywords:** video-gaming, cognitive empowerment, long-term effects, short-term effects, flow theory, automatization theory

## Abstract

The aim of the present study is to compare the short- and long-term effects of video-gaming by using the same measurements. More precisely, habitual and occasional video-gamers were compared so as to analyze the long-term effects. An ABABABA design was used to analyze the short-term effects. The first A refers to baseline measurements: Visual RT, Auditory RT, Aim trainer RT, Go/No-Go RT and N-Back RT. The first B refers to 30 min of gaming, the second A refers to the measurements used in the baseline, the second B refers to 60 min of a video game, the third A refers to the same measurements used in the baseline, the third B refers to a 30-min rest, and finally, the fourth A refers to the measurements used in the baseline. Seventy participants, twenty-nine habitual video-gamers and forty-one occasional video-gamers, participated in the study. The results showed a temporary improvement of cognitive functions (Visual RT, Auditory RT, Aim trainer RT, Go/No-Go RT and N-Back RT) in the short term and a strong enhancement of cognitive functions in the long term. The results are discussed in light of Flow Theory and the automatization process. Contribution of the study: The contribution of this research is to highlight that despite there being a transient enhancement of executive and cognitive functions through the use of mobile video games in the short-term period, with a decrease of performance after a 30-min rest, there is a strong increase of cognitive performance in the long-term period. Flow Theory and the automatization process together can explain this apparent inconsistency between the positive increase of long-term performance and the transient increase of short-term performance. One limitation of the present research is that it is not possible to distinguish whether the long-term enhancements can be attributed either to continued practice in the use of video games compared to non-gamers, or to the possibility that gamers are already predisposed to video game playing. Future research should address this issue.

## 1. Introduction

From a cognitive point of view, it is important to study both the long-term impact of video game playing and the short-term impact through intervention studies. To study the long-term impact, normally the cognitive profile of habitual and occasional video game players is analyzed. To study the short-term impact, the possibility of causally inducing changes in cognition via playing action video games is analyzed. Meta-analytical data converge to consistently demonstrate that video game players outperform non-video game players in a wide range of cognitive tasks [1,2,3,4,5,6]. The most recent meta-analyses on cognitive interventions report that with reference to different cognitive processes in healthy adults, video game training leads to cognitive improvement [6]. With reference to the long-term impact, habitual video game players show a better range of cognitive skills that specifically involve analogy, processing speed, deductive reasoning and mathematical intelligence [7]. Recent studies [8,9] have focused on the positive aspects of video games, not only on the negative ones, such as risk factors and adverse experiences [10,11,12,13]. Such studies suggest that commercial video games have the potential for improving cognitive functions [8,9]. To focus on understanding the association between video games and cognitive functions, it is essential to evaluate the potential of the results on the use of commercial video games [14]. Choi et al. [14] underline that video games, whose purpose is the entertainment of players, have been positively associated with some cognitive functions such as attention and problem-solving skills. Despite some discrepancies between studies [15,16], however, the improvement of cognitive functions through video games was limited to the task or performance that required the same cognitive functions [11]. A confirmation of the results related to attention comes from Dye and Bavelier [17], who recognized that expert gamers show higher levels of attention skills than occasional gamers, above all in their spatial and temporal resolution skills. Not only attention processes are enhanced through video-gaming. Hubana [18] recently analyzed the influence of puzzle video games on logical reasoning, using the analytical hierarchy process for evaluating and ranking the analyzed video games. His results demonstrate that puzzle games have a significant influence on logical reasoning. Zhang et al. [19] also try to show the mechanisms underlying how habitual video-gamers generalize learning from trained to untrained contexts. The authors underline that it happens as a sort of “learning to learn”—this mechanism appears when information and/or skills gained from experience in one task (or set of tasks) permit individuals to learn new tasks faster.

With reference to the second issue, the possible short-term impact of causally inducing changes in cognition via playing action video games, recently, Kozhevnikov, Li, Wong, Obana and Amihai [20] focused on transient enhancements of cognitive processes after video-gaming (short-term empowerment after 30 min). Their results showed that the cognitive empowerment was transient and ended after 30 min of rest activity. The authors were critical about the causal role of video game exposure. They suggest that transient empowerment may be due to specific factors, such as arousal or “flow” [21,22,23] or “peak experiences” [24]. Flow is an experiential state that occurs during full-capacity engagement, in which an individual is performing at a level that is matched with the demands of the task. When in a flow state, an individual is able to access maximal potential and perform at full capacity, while perceiving an optimal level of challenge and arousal without sensed stress [23]. Some authors investigated whether there are methods that allow to expand the results from a flow transient state to long term [24]. Looi, Duta, Brem et al. [25] combined transcranial direct current stimulation (tDCS) and a mathematical video game to investigate whether their synergistic effects on transfer and long-term change would be greater than training alone. They wondered if some short training would show long-term effects. The results the authors obtained showed that the tDCS and video game group obtained a benefit and retained this benefit over a period of 2 months without further video game training and tDCS.

Studies on the short-term effect of video games provide us with some information, but there is a lack of consistent evidence that long-term exposure to video games improves performance on non-game measures [26]. Since video game exposure determines only a temporary boost, the aim of the present study is to understand what happens when the long- and short-term effects are examined in the same experiment and with the same measurements.

To analyze the short-term effects of video games, we exposed both habitual and occasional video game players to repeated measurements (ABABABA) of cognitive performance after video-gaming (30 min of a video game, plus 60 min of a video game) and after a 30-min rest to understand if the short-term empowerment obtained from the previous phases persisted.

To analyze the long-term effects of video game exposure, we decided to compare habitual video game players (i.e., habitual players for more than 10 h per week and for at least 10 years) with occasional video game players (who play once or twice per week). The underlying logic is that by repeating the same activity, as happens for habitual players, some processes can become automatized and actual performance can show this long-term automatization.

## 2. Method

### 2.1. Participants

The subjects who took part in the research come from different areas of Italy: from both the north and south, and 70 subjects (34 males and 36 females) between the ages of 20 and 32 participated in the research. The average age of the subjects was 25 years old (SD = 6.22). Participants were recruited through social networks and voluntarily participated in the study. All subjects are of Italian nationality.

Twenty-nine healthy right-handed participants were habitual video-gamers (fifteen females), while forty-one healthy right-handed participants were occasional video-gamers (twenty-one females). Of the habitual video-gamer participants, 15.8% spend more than 5 h playing video games each day, 57.9% spend between 4 and 5 h and the remaining 23.6% spend 2–3 h each day. Gamers have been playing on average for about 10 years. Of the occasional video-gamers, 72.7% spend between 0 and 1 h per week, and 27.3% spend between 1 and 2 h per week. The subjects declared that the devices most often used for gaming were the Smartphone (61%), consoles (Sony’s Play Station and Microsoft’s Xbox) (22%), PC Gaming (12.2%), and finally, with a tablet (4.9%). All data were collected in compliance with Italian legislation on the protection of personal data, thus respecting and safeguarding the privacy and anonymity of the participants (Legislative Decree 196/2003), and Art. 9 of the Deontological Code of Italian Psychologists. Informed consent was obtained from all participants, and they were also asked to sign a form for the processing of personal data after full explanation of the procedure. Participation in the experiment was voluntary. 

### 2.2. Procedure

The procedure used is described in Figure 1. Participants, one day before the test, were asked to install the apps for the following video games on their devices: “Call of Duty mobile” and “Brawl Stars”. This procedure was required to familiarize the subjects with the commands to be used. After familiarization, an ABABABA design was used. The first A refers to baseline measurements: Visual reaction time (RT), Auditory RT, Aim trainer RT, Go/No-Go RT and N-Back RT. The first B refers to 30 min of gaming, the second A refers to the measurements used in the baseline, the second B refers to 60 min of a video game, the third A refers to the same measurements used in the baseline, the third B refers to a 30-min rest, and finally, the fourth A refers to the measurements used in the baseline (Figure 1). The session began by connecting with the participants on the ZOOM platform and inviting them to complete the anamnestic questionnaire for the collection of general data. A battery of neuropsychological tests was then administered to evaluate attention, executive functions, working memory and visuospatial memory. This battery of tests was presented in the following order: Visual Reaction Time (VRT), Auditory Reaction Time (ART), Aim trainer, GO/NO-GO Visual reaction time and finally the Working memory test N-Back. They were then asked to play video games for 30 min, 15 min starting with the Call of Duty mobile video game and then the other 15 min with Brawl Stars. At the end of the 30 min, the entire test battery was administered again. Next, the participants were asked to play video games for 60 min, alternating the two games every fifteen minutes by first running Brawl Stars and then Call of Duty mobile, in that order. Subsequently, there was a further administration of the battery of neuropsychological tests, identical to the first one. At the end of administration, each participant was told to take a break of 30 min, where it was clearly indicated that in this period no video game activity, practicing sports or listening to activating music could be carried out. At the end, again, the neuropsychological test battery was administered for the last time. The control group did not have to install any games. Each participant of the group, similar to the experimental group, was contacted via the ZOOM platform to participate in the research. The control group was administered the same test battery as the experimental group, respecting the four phases and the same timing, but without the introduction and exposure to video games. They were then invited to read some news or listen to quiet music. In total, the average time taken for the explanation of the procedures, the administration and the collection of data for the whole experimental phase was 3 h and 30 min for each individual participating.

### 2.3. Instruments

#### 2.3.1. Visual Search Task

The test was performed using the Cognitive Fun! Site (https://new.cognitivefun.net/ accessed on 10 November 2021). During the visual search task, in the center of the white screen background, the participant was presented with a fixation cross with a red dot in the center that was followed after variable time intervals by a target stimulus, that is, a green circle. The subject was asked to concentrate on the fixation cross and press the “space bar” key as soon as possible once the green circle (target stimulus) appeared on the screen [27].

#### 2.3.2. Auditory Search Task

In a simple auditory search task, after variable time intervals, sound was played for 30 s to the participant through the speakers. The task was to press the spacebar as soon as the sound was presented. The tests were performed using the Cognitive Fun! site. During the auditory search task, in the center of the white screen background, the participant was presented with a fixation cross with a red dot in the center, which was followed after variable time intervals by a target stimulus, that is, a dull sound. The subject was asked to concentrate on the fixation cross and press the “space bar” key as soon as possible once the dull sound (target stimulus) was heard from the audio source [28].

#### 2.3.3. AIM Trainer Measurement

The AIM Trainer test consists of hitting a moving target in the shortest time possible. It is a tool used above all to train competitive players in being able to “catch” and “hit” the target in the shortest possible time. The test was performed using the Human Benchmark site. During AIM Trainer, the subject had to visualize a moving target in the center of the screen, lock it with the mouse and hit it as quickly as possible with each movement. The test ended when the subject hit all 30 targets.

#### 2.3.4. Go/No-Go VRT Measurement

The two-choice procedure and the Go/No-Go procedure use RT and accuracy as dependent variables. The two-choice task is the most widely used of all RT-based procedures within the field of cognitive psychology [29,30].

The test was performed using the Cognitive Fun! Site. In this test, there is a stimulus that must be answered and an inhibitory stimulus that must not be responded to. The test consists of 15 trials. Each test began with the presentation of a red dot in the center of a white background which had to be fixed for about 4 s. The request was to click the space bar when a solid green dot was displayed and ignore the dot with different shades of green and with a segmented texture. During the procedure, only the wrong answer came with a red cross. 

#### 2.3.5. Visual and Auditory Working Memory Measurement 

In order to evaluate the performance of working memory (WM), n-back was used, a continuous performance activity commonly adopted in the field of psychology and cognitive neuroscience. This paradigm has been widely used in the literature and has shown good psychometric properties [31,32,33]. The n-back task involves the serial presentation of stimuli (for example, an image or a sound), separated from each other by a few seconds. The participant has to decide whether the current stimulus matches that shown/heard in previous steps. n indicates the load factor, a variable number that can be adjusted up or down, respectively, to increase or decrease the cognitive load and to make the task more or less difficult [34]. This activity involves the active part of WM, as it requires the maintenance and continuous updating of information. In this study, the 2-back versions of both visual and auditory activity were used. It was explained to participants that, after beginning a succession of 50 images from the start, they had to click on the figure that appeared 2 positions before. For example, if the sequence was machine–heart–machine, it was necessary to click on this last image since it had appeared two positions before. Each image appeared on the screen for 500 ms, followed by a screen that remained blank for another 3000 ms. The participant had 3500 ms, from stimulus onset until the beginning of the subsequent trial, to press the space bar. The https://new.cognitivefun.net/website (accessed on 25 April 2020) was available for 175 s for each of the two tests. When introducing the auditory n-back test, it was explained to the participants that it was necessary to proceed similarly to the previous activity, but instead of the 50 images, they would hear a sequence of 50 sounds; then, they had to click the space bar when they heard the specific sound of two positions before. The average completion time for participants was from 120 to 175 s for the visual n-back and from 112 to 175 for auditory n-back tests.

### 2.4. Statistical Analysis 

Data were analyzed using SPSS Version 24.0 for Mac. Measurement parameters varied according to the type of task, means of correct responses (CR) and means of correct reaction times (RT) obtained for each participant, for each experimental condition. The parameters were submitted to 2 × 2 × 4 mixed analysis of variances with Group (experimental and control) and Video-gamer (habitual and occasional) as between-subjects factors, and Phase (1: baseline, 2: after 30 min of play, 3: after one hour of play and 4: after a 30-min break) as within-subject factors. Descriptive statistics of the dependent variables were tabulated and examined. RT data referring to correct replies were cleaned, and outliers were removed (outliers referred to more than 1000 ms or less than 200 ms, and accounted for 2% of responses). Alpha level was set to 0.05 for all statistical tests. In the case of significant effects, the effect size of the test was reported. For ANOVA, partial eta-squared (pη^2^) was used, and for the *t*-test, Cohen’s d Effect Size was used. The Greenhouse–Geisser adjustment for non-sphericity was applied to probability values for repeated measures.

## 3. Results

Results are described below in relation to the research questions posed. With reference to the first question, the short-term effects of video games, we analyzed both habitual and occasional video game players exposed to repeated measurements (ABABABA) of cognitive performance to understand if the short-term empowerment obtained in the first three A phases persisted after resting.

With reference to the Visual Search task, RT was considered, and Table A1 (Appendix A) shows the means and standard deviations of each phase for each group. Phase shows significant effects, F (3, 198) = 4.6, *p* < 0.004, pη^2^ = 0.08. The Group X Phase interaction was also statistically significant, F (3, 198) = 3.78, *p* < 0.01, pη^2^ = 0.08. The RT of the control group exhibited a stable performance over time, while the experimental group decreased RT after video-gaming and increased RT after the break phase. These data may indicate that there is a short-term activation induced by exposure to video games, however this activation is not maintained over time after the subject has finished the break phase (Figure 2).

Post-hoc analysis with paired *t*-test showed that phase 1 was statistically different compared to phase 2 and phase 3, respectively, *t* (40) = 2.63, *p* < 0.01, d = 0.77, and *t* (40) = 4.16, *p* < 0.0001, d = 0.99, but it was not statically different compared to phase 4, *t* (40) = 1.15, *p* < 0.09, d = 0.55. Phase 2 was not statistically significant compared to phases 3 and 4. Comparing phase 3 and phase 4, the *t*-test was again significant, *t* (29) = 3.30, *p* < 0.002, d = 0.74. With reference to the Auditory Search task, RT was considered, and Table A2 (Appendix A) shows the means and standard deviations of each phase for each group. Neither Phase nor Group showed significant effects. With regard to the Aim Trainer test, RT was considered, and Table A3 (Appendix A) shows the means and standard deviations of each phase for each group. Neither Phase nor Group showed significant effects.

With reference to the Go/No Go task, Table A4 (Appendix A) presents the means and standard deviations of each phase for each group. The Group X Phase interaction was statistically significant, F (3, 198) = 2.28, *p* < 0.04, pη^2^ = 0.08, and the RT of the control group exhibited a stable performance over time, while the experimental group decreased RT after video-gaming and increased RT after the break phase. These data may indicate, again, that there is a short-term activation induced by exposure to video games, however this activation is not maintained over time after the subject has finished the break phase (Figure 3). 

With regard to the Working Memory N-Back test, Table A5 (Appendix A) presents the means and standard deviations of each phase for each group. The Phase factor showed significant effects, F (3, 195) = 12.92, *p* < 0.0001, pη^2^ = 0.075, and the Phase X Group interaction also showed significant effects, F (3, 195) = 3.44, *p* < 0.01, pη^2^ = 0.068. 

As can be seen in Figure 4, both groups improved their performances, however, improvement in the experimental group was higher than improvement in the control group. There was no increase in RT after the break, as in the previous results. 

Post-hoc analysis with the paired *t*-test showed that phase 1 was statistically different compared to phase 2 and phase 3, respectively, *t* (29) = 3.97, *p* < 0.0001, d = 0.77, and *t* (29) = 5.57, *p* < 0.0001, d = 0.99, but it was not statistically different compared to phase 4, *t* (40) = 1.15, *p* < 0.09, d = 0.55. Phase 2 was not statistically significant compared to phases 3 and 4. Comparing phase 3 and phase 4, the *t*-test was again not significant. These data confirm that there was a significant decrease in RT in the N-Back Working Memory test already after playing for half an hour, and this decrease was stable. 

With reference to the second question, the long-term effects of video games, we compared habitual video game players with occasional video game players. With reference to the Visual Search task, the factor Video-gamer (habitual vs. occasional) showed significant effects, F (1, 66) = 18.42, *p* < 0.0001, pη^2^ = 0.09. This means that habitual video-gamers have faster reaction times in Visual Search than occasional video-gamers. With reference to the Auditory Search task, the factor Video-gamer again showed significant effects, F (1, 66) = 13.05, *p* < 0.001, pη^2^ = 0.11. This again means that habitual video-gamers have faster reaction times in Auditory Search than occasional video-gamers. With regard to the Aim Trainer test, the factor Video-gamer showed significant effects, F (1, 66) = 6.2, *p* < 0.01, pη^2^ = 0.09. This means that habitual video-gamers have faster reaction times in the Aim Trainer task than occasional video-gamers, with this also being one of the basic activities in the interaction of shooter video games, such as the one used in this work. Finally, with reference to the Go/No Go task and n-back task, the factor Video-gamer showed significant effects, respectively, F (1, 66) = 9.39, *p* < 0.003, pη^2^ = 0.082, and F (1, 66) = 12.92, *p* < 0.0001, pη^2^ = 0.075. This means that even with reference to more complex cognitive processes, video-gamers tend to have better performances than occasional video-gamers. 

## 4. Discussion

With this study, we aimed to shed light on the association between the long- and short-term effects of video gaming, examined in the same experiment and with the same measurements. Data from the present research show that there is enhancement of attention and executive functions in the short term for gamers and non-gamers of the experimental group. These findings are in line with those of prior works [18,19,20,21], which showed that some cognitive processes, such as attention and perception, improved for short periods of time in subjects exposed to activating stimuli such as video games. Green and Bavalier [5] also suggested that some types of video games such as action games (e.g., the Call of Duty game used in this research) appear to improve performance in tasks involving visual attention [35,36,37,38]. In agreement with Kozhevnikov, Li, Wong, Obana and Amihai [20], we think that this increase may be due to flow experience, the ability to access maximal potential and perform cognitive tasks at full capacity, while perceiving an optimal level of challenge and arousal without sensed stress [10,39]. Taking the above study as a reference point, where the authors observed that the cognitive improvements obtained after 30 min of rest after playing video games ceased, in this study there was also a decrease in performance enhancement in all cognitive tests, with the exception of the Working Memory test. We were surprised to find that there was no significant improvement of RT in game-players’ performance on the n-back memory task after 30 min of rest. It is possible that the enhanced cognitive states experienced by the players in these experiments lasted more than 30 min following gaming. Furthermore, it could be that the temporal characteristics of memory, as measured by the n-back task, decline more slowly than the other tasks related to attention measures. Another possibility, however, is that learning during the enhanced state was especially efficient, enabling players to develop potent strategies for performing the n-back task. 

With reference to the second question posed, the long-term effect, the results of the present research show that gamers are better in terms of performance in all cognitive tests than non-gamers, and these findings are in line with those of prior works [21].

Now, the open question is: how do the results that appear to be transient show strong effects on long-term performances (with habitual video-gamers)? It may be that habitual video-gamers are working on their routine exercises [40,41,42,43,44]. Possession of expertise permits well-organized retrieval and coding of information, and the learning and automatization of game-play abilities provide them with essential cognitive processes for enabling the transfer of higher cognitive processes. Automatization allows the possibility to process information with minimal cognitive effort (working memory), and consequently, freeing working memory sources works [19,43,44]. The occasional player has not automatized some cognitive processes and can accomplish cognitive tasks through a slow and awkward process. As Hubana [18] underlined, increased attention control could be one of the mechanisms through which improved automatization occurs. In this view, attentional control processes, which encompass cognitive flexibility and working memory, act as a guide to identify and to keep track of task-relevant features, and thus facilitate learning in other settings. 

## 5. Conclusions

The results of the present study are two-fold: they demonstrate that there is an enhancement of cognitive functions through the use of mobile video games in the short term, and this implies the existence of temporary cognitive states in which auditory and visual aspects of attention are dramatically enhanced for limited durations. This is similar to what happens in flow experiences. With reference to the long-term effect, habitual video-gamers show better cognitive performances than occasional video-gamers. In summary, the present work documented two pathways for cognitive training: the first acts in the short term and temporarily amplifies cognitive performances, whereas the second takes place in the long term and strongly increments cognitive performances. 

### 5.1. Limitations and Future Research

One limitation of the present research is that it is not possible to distinguish from the results if the enhancements of cognitive performances can be attributed either to a lasting improvement due to continued practice in the use of video games compared to non-gamers, or to the possibility that gamers are already predisposed (e.g., have more activated arousal levels) and thus more likely to be activated. Consequently, when subjected for the first time to the video game world, they become gamers who are capable of better basic performance than those who do not have these characteristics, and this could create self-reinforcement that works as a virtuous circle that in turn makes further progress possible. This first limitation could be a starting point to be explored for expanding this research, to then study what the variables are that identify gamers and their ability to develop better skills of executive functions than other non-gaming subjects. Future research should examine groups of video-gamers with different years of experience (for example 1, 2, 3 and so on) to identify when the cognitive abilities show these enrichments. The complete range of attentional capacities might also be analyzed through diverse attention tasks: general executive and distributed attention tasks. Assessing attention might further examine certain aspects in detail, for instance, how long empowerment lasts. Furthermore, factors also need to be considered in detail, including baseline attentional capacities, perception of challenge during the activity and overall experience of the activity. Finally, it would be interesting to understand how to best reach efficient learning levels and, since arousal seems to be a critical prerequisite for reaching enhanced cognitive states, future studies should examine the physiological measures of arousal and how to heighten it. Since Looi et al. [25] have shown that tDCS and video games together obtained a benefit and retained this benefit over a period of two months without further video game training and tDCS, future research should further study this aspect. It may be that tDCS stimulates a faster automatization process.

Finally, we found several works that addressed issues such as motivation and preference related to video games, showing that younger players prefer action games, and older players games of skill [45,46,47]. Mixed results emerged concerning age differences with reference to video games [48,49,50], but younger gamers especially seemed to be motivated for video gaming by social interactions [51]. Nothing is known on the effects of the age of video-gamers on cognitive performances. In the present study, we use a sample with a mean age of 25 years. Does the fact that video-gamers spend a lot of time playing video games render them independent from age in cognitive performances? Future research should also address this issue.

### 5.2. Practical Implications

Apart from theoretical findings, there were also four practical findings. Firstly, video-gaming is put forward as a way for neuroscientists and cognitive psychologists to examine enhanced states. Secondly, the implications for the general population could be very interesting. Albeit brief, focused attention, no matter how short, could be of benefit in critical moments. Thirdly, the possibility that creativity may be associated with an enhanced cognitive state [39] is an interesting path for future studies. Fourth, and finally, while these enhanced states may be transitory, it could be that the learning achieved is long lasting. 

Undoubtedly, enhanced states need further investigation, along with how to stimulate them. Retrieving dormant cognitive capacities and enhancing them at will could thus be greatly increased.

## Figures and Tables

**Figure 1 ijerph-19-00078-f001:**
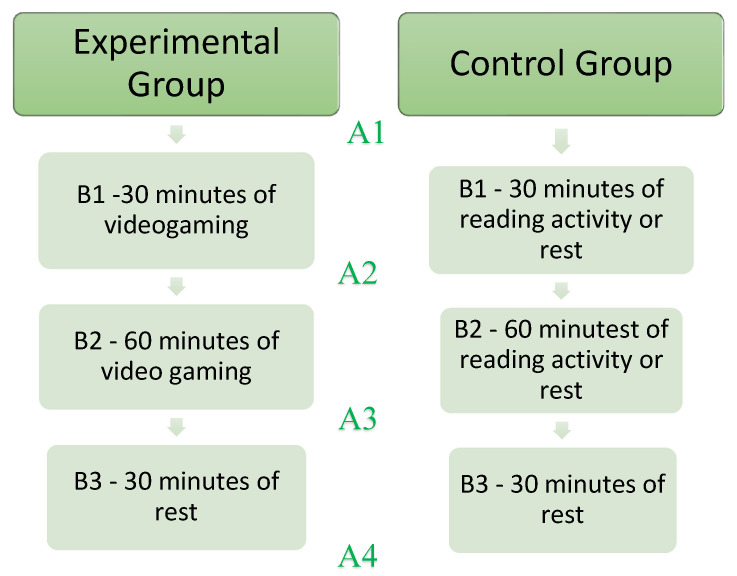
Research plane. Each A refers to each of the four measurement phases, each B refers to the intervention phases.

**Figure 2 ijerph-19-00078-f002:**
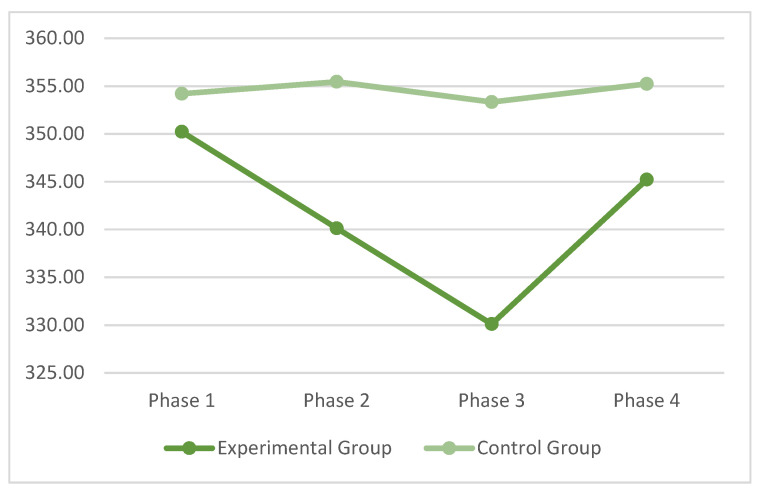
RT means in the four phases of the Visual Search Task.

**Figure 3 ijerph-19-00078-f003:**
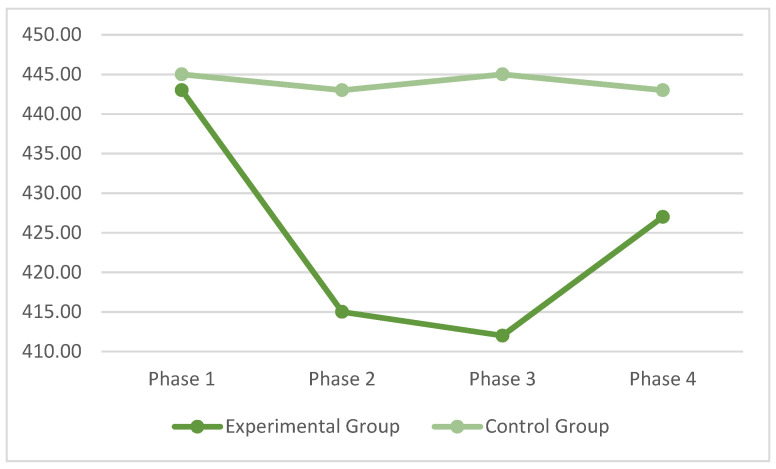
Means of the RT in the four phases of the Go/No Go Visual Research task.

**Figure 4 ijerph-19-00078-f004:**
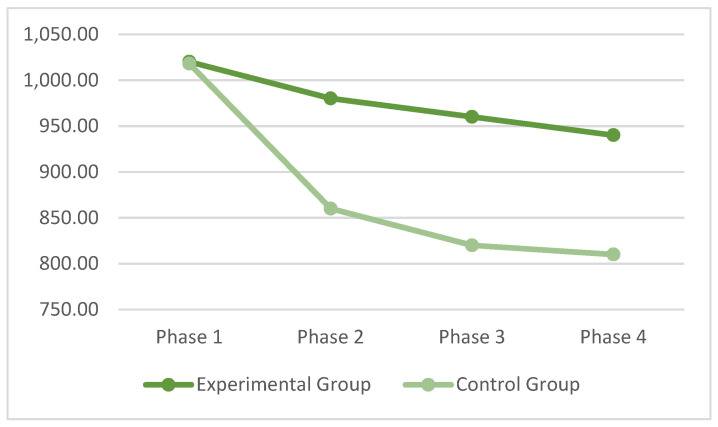
Means of the RT in the four phases of the N-Back task.

## Data Availability

Dataset is available from the link: https://data.mendeley.com/datasets/p9p8w5k2cc/1, accessed on 10 November 2021.

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
