# Peer review of "Transient and Long-Term Improvements in Cognitive Processes following Video Games: An Italian Cross-Sectional Study"

_ijerph, 2021, doi:10.3390/ijerph19010078_

Round 1
Reviewer 1 Report
The article corresponds to the profile of a scientific journal. However, the issue of the elaboration of the topic is not fully presented. For example, the question about Combining brain stimulation and video game to promote long-term transfer.
The authors should be more clear on the methodology. The gender composition of gamers and non-gamers is not clear — their age component.
The way young people solve tests at 20 differs from how they do it at 32. This difference is due to objective changes, and therefore, these issues must also be taken into account.
Author Response
- The article corresponds to the profile of a scientific journal. However, the issue of the elaboration of the topic is not fully presented. For example, the question about combining brain stimulation and video game to promote long-term transfer.
Reply
Thank you. We added information as indicated on both issues in the introduction and in the conclusion sections.
- The authors should be clearer on the methodology. The gender composition of gamers and non-gamers is not clear — their age component.
Reply:
Thank you, we added the gender composition of gamers and non-gamers, and an indication regarding age. We highlighted the text in green.
- The way young people solve tests at 20 differs from how they do it at 32. This difference is due to objective changes, and therefore, these issues must also be taken into account.
Reply:
Thank you for your suggestion. We found several works that addressed this issue: most of them refer to the preferred types of video games and to the motivation showing that younger players prefer action games, older players games of skill (Nap et al., 2009; Scharkow et al., 2015; Homer et al., 2012; Mihara and Higuchi, 2017). Mixed results emerged concerning age differences with reference to video games (Greenberg et al., 2010), but younger gamers especially seemed to be motivated for video gaming by social interactions (Von der Heiden, et al., 2019; Hilgard et al., 2013). We added these issues in both the limitations and in the future direction sections.
Reviewer 2 Report
Manuscript ID ijerph-1482285, titled: " Transient and long-term enhancements of cognitive processes after videogaming." In this work, 70 participants, 29 regular video gamers and 41 occasional video gamers, participated in the study. To analyze the long-term effects, regular and occasional video gamers were compared. The results showed a temporary improvement in short-term executive and cognitive functions and a significant improvement in long-term cognitive functions.
To improve your work, I suggest:
- Check the title; it seems to be a cross-sectional study applied to participants from Italy. The title could be "Transient and long-term improvements in cognitive processes following video games in Italy: A cross-sectional study".
- In the Abstract, include a paragraph of the contribution of this research, limitations and future work.
- Check the English grammar and spelling of the entire paper.
- Indicate whether this research uses publicly available information or has a data set with access to the data for researchers to replicate the experiment.
- The article contains many tables; I suggest summarizing the most relevant data or findings.
- If you wish to keep the tables, I suggest placing them as an appendix.
- To make your research replicable, I suggest placing the study data in a dataset; you can create one at https://data.mendeley.com/.
- Improve the quality and resolution of Figure 1a.
- Improve the quality and resolution of Figure 1b.
- Place the tables in the format suggested by the magazine, not as images.
- Explain the method better; if possible, include a sketch of the applied method.
- Improve the discussion section.
- Improve the conclusions section.
- I suggest updating the references 62.5% are outside the five years of validity.
- I encourage them to review other similar works to improve their article, for example, https://www.mdpi.com/1660-4601/18/10/5063
https://www.mdpi.com/1660-4601/18/10/5474
https://www.mdpi.com/1660-4601/18/10/5492
Author Response
- The article corresponds to the profile of a scientific journal. However, the issue of the elaboration of the topic is not fully presented. For example, the question about combining brain stimulation and video game to promote long-term transfer.
Reply
Thank you. We added information as indicated on both issues in the introduction and in the conclusion sections.
- The authors should be clearer on the methodology. The gender composition of gamers and non-gamers is not clear — their age component.
Reply:
Thank you, we added the gender composition of gamers and non-gamers, and an indication regarding age. We highlighted the text in green.
- The way young people solve tests at 20 differs from how they do it at 32. This difference is due to objective changes, and therefore, these issues must also be taken into account.
Reply:
Thank you for your suggestion. We found several works that addressed this issue: most of them refer to the preferred types of video games and to the motivation showing that younger players prefer action games, older players games of skill (Nap et al., 2009; Scharkow et al., 2015; Homer et al., 2012; Mihara and Higuchi, 2017). Mixed results emerged concerning age differences with reference to video games (Greenberg et al., 2010), but younger gamers especially seemed to be motivated for video gaming by social interactions (Von der Heiden, et al., 2019; Hilgard et al., 2013). We added these issues in both the limitations and in the future direction sections.
Referee 2
- Manuscript ID ijerph-1482285, titled: " Transient and long-term enhancements of cognitive processes after videogaming." In this work, 70 participants, 29 regular video gamers and 41 occasional video gamers, participated in the study. To analyze the long-term effects, regular and occasional video gamers were compared. The results showed a temporary improvement in short-term executive and cognitive functions and a significant improvement in long-term cognitive functions.
To improve your work, I suggest:
Check the title; it seems to be a cross-sectional study applied to participants from Italy. The title could be "Transient and long-term improvements in cognitive processes following video games in Italy: A cross-sectional study".
Reply
Thank you for your suggestion: we changed the title (in green) as follows:
Transient and long-term improvements in cognitive processes following video games: An Italian cross-sectional study
- In the Abstract, include a paragraph of the contribution of this research, limitations and future work.
Reply
We rewrote the abstract and inserted these suggestions.
- Check the English grammar and spelling of the entire paper.
Reply
A native English speaker thoroughly revised the work and corrected it.
- The article contains many tables; I suggest summarizing the most relevant data or findings.
If you wish to keep the tables, I suggest placing them as an appendix.
Reply
Thank you. We added the tables as an appendix
- Indicate whether this research uses publicly available information or has a data set with access to the data for researchers to replicate the experiment.
To make your research replicable, I suggest placing the study data in a dataset; you can
create one at https://data.mendeley.com/.
Reply
Thank you. We inserted our data on the site you suggested us. The link is:
https://data.mendeley.com/datasets/p9p8w5k2cc/1
we added in the paper that the data set is accessible at the above link.
- Improve the quality and resolution of Figure 1a.
- Improve the quality and resolution of Figure 1b
Reply
Thanks. We redid Figure 1 and 2 (summarizing them into a single figure.
- Place the tables in the format suggested by the magazine, not as images.
Reply
We did this.
- Explain the method better; if possible, include a sketch of the applied method.
Reply
Thank you. We explained the method better by introducing the ABABABA design and
figure 1.
- Improve the discussion section.
- Improve the conclusions section.
Reply
Both sections have been improved
- I suggest updating the references 62.5% are outside the five years of validity.
I encourage them to review other similar works to improve their article, for example, https://www.mdpi.com/1660-4601/18/10/5063
https://www.mdpi.com/1660-4601/18/10/5474
https://www.mdpi.com/1660-4601/18/10/5492
Reply
Thank you. We did it
Reviewer 3 Report
I have read manuscript with great attention and interest. The article deals with an interesting topic of gaming impact to some attributed of people. This topic is quite needed. The topic is interesting but the quality of the research must be improved to suit the high impact-factor journal.
Comments and suggestions:
- Introduction is quite short. Please describe similar research extensively and compare it with your one.
- Why is the research of Rubinstein highlighted?
- Your tables are pasted as figures. This should be corrected and they should be pasted as text.
- The article is hard to read and hard to follow your ideas. Please edit it with more experienced English write.
I recommend to improve the article to suit the high impact-factor journal.
Author Response
I have read manuscript with great attention and interest. The article deals with an interesting topic of gaming impact to some attributed of people. This topic is quite needed. The topic is interesting but the quality of the research must be improved to suit the high impact-factor journal.
Comments and suggestions:
- Introduction is quite short. Please describe similar research extensively and compare it with your one.
Reply
Thank you. We rewrote the introduction and added new research.
- Why is the research of Rubinstein highlighted?
Reply
You are right. Rubinstein’s research is just one of many authors that
used the Go/No- Go task with the two-choice procedure; we added more recent and
significant research such as Gomez et al. (2009) as he and his collaborators did
an accurate review of this method.
- Your tables are pasted as figures. This should be corrected and they should be pasted as text.
Reply
Done
- The article is hard to read and hard to follow your ideas. Please edit it with more experienced English write.
Reply
Thank you. The article has been revised by a native English speaker
- I recommend to improve the article to suit the high impact-factor journal.
Reply
Thank you, we rewrote much of the article and extended it, and improved it so it is now suitable for a high impact-factor journal.
Reviewer 4 Report
The authors address an interesting topic related to the use of video games and their implications for cognitive processes.
Although there are aspects that need to be improved:
- The abstarct should be redone, an abstact should include the objectives of the research and aspects related to the method in which data on the sample should be included, and simple aspects of the procedure, the most representative tests used and the most significant results found. The abstarct currently includes some but not all of these aspects.
- The instruction, i.e. the theoretical justification is too short and should be expanded as it is a very topical subject and there is a lot of information on the subject. The authors only include 30 references, the number of references should be expanded and updated, as only 33.33% are references to articles in the last five years, at least 60% of the total number of updated citations should be presented.
- The research questions should be clearly stated at the end of the introduction section.
- Citations in the manuscript do not follow the journal's rules, they should not be presented with the author's name and date in brackets, but should be numbered and in square brackets.
- In the method section, under the sub-section on participants, it should be clearly stated that consent was informed and in writing.
- In the results section, authors should describe the results found in relation to the research questions posed. They should also carefully review the journal's guidelines for including tables and figures, as they do not follow the current format of the journal's guidelines.
- The journal template includes the discussion sections that the authors have not addressed, in which they should relate the results found with previous studies that have served as a basis for the state of the art and analyse the coincidences or non-coincidences and argue why. It is recommended that they review the journal's template.
- The conclusions section should reflect the most significant findings of the work, the limitations and future lines of research.
- It is also recommended that authors follow the journal template and address the following sections Funding and Acknowledgments.
- Citations in the references section do not follow the journal's guidelines, it is recommended that they be revised and presented in the format established by the publisher.
Author Response
The authors address an interesting topic related to the use of video games and their implications for cognitive processes.
Although there are aspects that need to be improved:
- The abstract should be redone, an abstract should include the objectives of the research and aspects related to the method in which data on the sample should be included, and simple aspects of the procedure, the most representative tests used and the most significant results found. The abstract currently includes some but not all of these aspects.
Reply
We rewrote the abstract
- The instruction, i.e. the theoretical justification is too short and should be expanded as it is a very topical subject and there is a lot of information on the subject. The authors only include 30 references, the number of references should be expanded and updated, as only 33.33% are references to articles in the last five years, at least 60% of the total number of updated citations should be presented.
Reply
Thank you. We have enriched the text (in green) and included more recent
articles.
- The research questions should be clearly stated at the end of the introduction section.
Reply
We have done this.
- Citations in the manuscript do not follow the journal's rules, they should not be presented with the author's name and date in brackets, but should be numbered and in square brackets.
Reply
This has been corrected throughout the manuscript.
- In the method section, under the sub-section on participants, it should be clearly stated that consent was informed and in writing.
Reply
Thank you. We rewrote it in the suggested section.
- In the results section, authors should describe the results found in relation to the research questions posed. They should also carefully review the journal's guidelines for including tables and figures, as they do not follow the current format of the journal's guidelines.
Reply
We described them in relation to question posed, as suggested. We have also followed
the journal’s guidelines regarding tables and figures.
- The journal template includes the discussion sections that the authors have not addressed, in which they should relate the results found with previous studies that have served as a basis for the state of the art and analyse the coincidences or non-coincidences and argue why. It is recommended that they review the journal's template.
Reply
We included a discussion section
- The conclusions section should reflect the most significant findings of the work, the limitations and future lines of research.
Reply
We added this part in the conclusion section
- It is also recommended that authors follow the journal template and address the following sections Funding and Acknowledgments.
Reply
We followed it and included Funding, but we have no acknowledgments to add.
- Citations in the references section do not follow the journal's guidelines, it is recommended that they be revised and presented in the format established by the publisher.
Reply
Thank you. We rewrote the citations
Round 2
Reviewer 1 Report
The authors have tried to take into account all the comments made earlier.
Author Response
thank you
Reviewer 2 Report
With the changes applied to the document it has improved significantly. I suggest it be considered for publication.
Author Response
Thank you
Reviewer 3 Report
I think that the article can be accepted.
Author Response
thank you
Reviewer 4 Report
The authors have made the changes suggested in the review. Although the abstract should be improved, it is too long; normally 150 to 250 words are recommended. Also, the figures should be improved, specifically figure 1. Likewise, the citation in the bibliographical references section should be revised.
Author Response
Dea editor, thank you.
The following corrections have been made to the study.
The authors have made the changes suggested in the review.
Although the abstract should be improved, it is too long; normally 150 to 250 words are recommended.
Reply
Thank you. We deleted some text and we wrote only the essential details. Moreover, we moved on another section your requested “contribution of the study”.
Also, the figures should be improved, specifically figure 1.
Reply
We revised the figure 1
Likewise, the citation in the bibliographical references section should be revised.
Reply
Thanks. We revised it.